# Enzymatic Protein Biopolymers as a Tool to Synthetize Eukaryotic Messenger Ribonucleic Acid (mRNA) with Uses in Vaccination, Immunotherapy and Nanotechnology

**DOI:** 10.3390/polym12081633

**Published:** 2020-07-23

**Authors:** Fabiola Urbina, Sebastián Morales-Pison, Edio Maldonado

**Affiliations:** 1Programa de Biología Celular y Molecular, Instituto de Ciencias Biomédicas, Facultad de Medicina, Universidad de Chile, Independencia 1027, Santiago 8380453, Chile; fabi.urbina1516@gmail.com; 2Laboratorio de Genética Molecular Humana, Programa de Genética Humana, Instituto de Ciencias Biomédicas, Facultad de Medicina, Universidad de Chile, Independencia 1027, Santiago 8380453, Chile; seba.morales.p@gmail.com

**Keywords:** protein biopolymer, mRNA, transcription, vaccine, immunotherapy, nanotechnology

## Abstract

Multi-subunit enzymes are protein biopolymers that are involved in many cellular processes. The enzyme that carries out the process of transcription of mRNAs is RNA polymerase II (RNAPII), which is a multi-subunit enzyme in eukaryotes. This protein biopolymer starts the transcription from specific sites and is positioned by transcription factors, which form a preinitiation complex (PIC) on gene promoters. To recognize and position the RNAPII and the transcription factors on the gene promoters are needed specific DNA sequences in the gene promoters, which are named promoter elements. Those gene promoter elements can vary and therefore several kinds of promoters exist, however, it appears that all promoters can use a similar pathway for PIC formation. Those pathways are discussed in this review. The in vitro transcribed mRNA can be used as vaccines to fight infectious diseases, e.g., in immunotherapy against cancer and in nanotechnology to deliver mRNA for a missing protein into the cell. We have outlined a procedure to produce an mRNA vaccine against the SARS-CoV-2 virus, which is the causing agent of the big pandemic, COVID-19, affecting human beings all over the world. The potential advantages of using eukaryotic RNAPII to synthetize large transcripts are outlined and discussed. In addition, we suggest a method to cap the mRNA at the 5′ terminus by using enzymes, which might be more effective than cap analogs. Finally, we suggest the construction of a future multi-talented RNAPII, which would be able to synthetize large mRNA and cap them in the test tube.

## 1. Introduction

Biopolymers are molecules produced by living organisms, which contain monomeric units that are covalently linked to form larger structures. There are three classes of biopolymers, classified according to the monomeric units used to form the structure of the biopolymer; first, the polynucleotides (DNA and RNA), composed by nucleotides; second, the proteins formed by amino acid residues covalently bound by peptide bonds; and third, the polysaccharides, which are often linear bonded polymeric carbohydrate structures.

The DNA molecule is the most important biopolymer carrying all the genetic instructions for the development, functioning, growth, and reproduction of all known organisms. In addition, all cellular organisms use messenger RNA (mRNA) to convey genetic information that direct synthesis of specific proteins. Proteins are biopolymers, which are able to perform most of the cellular functions in an organism.

Essentially, transcription of eukaryotic mRNA is the process of copying an RNA, using it as a template of the DNA, which is carried out by the enzymatic protein biopolymer RNA polymerase II (RNAPII). The template DNA is a double helix and the RNAPII transcription machinery must recognize the promoter element on the coding strand (or the mRNA-like strand) and copies the mRNA on the template strand (Figure 1). The transcription process is mainly carried out in three steps, which are initiation, elongation, and termination [1].

In this review, we will focus on the function of the eukaryotic enzyme RNAPII, which is a complex protein biopolymer able to synthetize the mRNA and the functions of additional protein factors required by this enzyme in the process of transcription. We will also focus on the design and uses of mRNA-based vaccines, mRNA nanomedicines and on the advantages of using eukaryotic RNAPII to synthetize functional mRNA. We proposed an idea of how to make a poly-talented eukaryotic RNAPII and a chimeric capping system. 

## 2. Proteins and Core DNA Promoter Elements Involved in the Transcription Process

The core of the transcription machinery to synthetize the mRNA is the complex protein biopolymer RNAPII, which is composed of about 12 to 14 polypeptides subunits (named Rpb) held together mainly by strong hydrophobic interactions [2]. The X-ray crystallographic structure of budding yeast RNAPII was solved and revealed four mobile elements termed Core, Clamp, Shelf, and Jaw Lobe [3]. Although all the four mobile elements are important for the transcription process, perhaps one of the more important is a region in which Rpb1 and Rpb2 form the active center of the enzyme [3]. In addition, we need to mention another domain contained in the Rpb1 subunit, which is named the C-terminal domain (CTD) and which is a hallmark of eukaryotic RNAPII [4]. The CTD is able to act as a platform for the assembly of factors involved in the different transcription steps, such as initiation, activation, elongation, termination (including 3′ polyadenylation), and mRNA processing, which includes 5′ capping and splicing [5]. 

The transcription initiation process carried out by core RNAPII requires an additional set of protein factors [1]. That set of protein factors, named General Transcription Factors (named TFIIs and GTFs), includes the TFIIA (Transcription Factor IIA), TFIIB (Transcription Factor IIB), TBP (TATA binding protein), TFIIE (Transcription Factor IIE), TFIIF (Transcription Factor IIF), and TFIIH (Transcription Factor IIH) (Table 1) [1]. Although they do not form part of the core RNAPII itself, they do perform essential functions during the transcription initiation process and during the transcription elongation step. One of the first steps during the transcription initiation process of mRNA is the recognition of a sequence element in the gene promoter by a TFII [6]. There are several sequence elements found in core promoter elements including the TATA element (TBP binding site), BRE (B recognition element), INR (initiator element), DPE (downstream promoter element), and the Homology D box element (HomolD-box) found in all promoters of ribosomal proteins encoding genes (RPGs) in fission yeast [6,7,8]. Afterwards, we will compare the promoter recognition of the TATA element and the HomolD-box element of the RPGs. Once a TFII is able to recognize a promoter element by using a low affinity DNA-protein interaction, several other ordered protein-protein interactions are produced in between the TFIIs to bridge and position the core RNAPII to produce a preinitiation complex (PIC), which is ready to initiate transcription when all four ribonucleotides triphosphate are added [1,8]. This PIC is a large protein biopolymer with a molecular weight of about 3 M Da [1]. During the last decade, the X-ray crystallographic structures of several PICs containing different arrangement of RNAPII, TFIIs (GTFs) and coactivators have been solved and they are reviewed in reference [9].

The classical pathway described of PIC formation is on core promoters that contain a TATA element and are described in Figure 2. Core promoters containing a TATA element are bound by TBP, which follows the binding of TFIIB [1]. Once TFIIB is incorporated into the complex, it follows the binding of core RNAPII-TFIIF, followed by TFIIE and TFIIH, respectively (Figure 2) [1]. For those core promoters which do not contain a TATA element (almost 70%), for example those containing an INR element, TBP in combination with another TFII could be able to recognize this element (Figure 3) (Maldonado, Edio; unpublished results). Afterwards, the PIC formation is identical to those containing a TATA element (Figure 3). The gene promoters of RPGs in fission yeast contain a Homology D box element and this element is recognized by a protein factor named Rrn7, which is related to TFIIB. Once that this protein factor recognizes the HomolD-box element, a dimer TPB-TFIIB is able to recognize it. Afterwards, the pathway of PIC formation is exactly as the PIC formation on TATA-containing promoters (Figure 4) [10].

The PIC formed on those promoters containing TATA, INR or HomolD-box elements in which only TFIIs and core RNAPII are involved can only specify basal or non-regulated transcription. Transcriptional activation carried out by specific regulatory factors that bind specific sequences upstream of the transcriptional start site often needs another multiprotein complex, besides the TFIIs and core RNAPII. This complex is named the mediator (Table 1) and was first identified in yeast and soon after in higher eukaryotes [11,12]. In yeast, it is composed of about 24 subunits called Meds, which can serve as regulatory factor targets [1,11,12]. In agreement with this notion, the Med subunits can bridge the specific regulatory factors with the PIC composed by the TFIIs and core RNAPII [1,11,12].

## 3. mRNA as a Tool for RNA-Based Vaccines

The conventional vaccination is one of the major breakthrough in modern Medicine, which have reduced the incidence of infectious diseases such as measles and completely eradicating others such as smallpox. However, conventional vaccination has not been effective against another infectious diseases such as influenza or other emerging diseases such as those caused by Zika and Ebola viruses. Nowadays, we are facing a devasting pandemic all around the world caused by the SARS-CoV-2 virus which causes the COVID-19 disease. This pandemic has caused millions of infected people and several thousands of deaths. Additionally, this pandemic is devasting the economies of most of the countries. mRNA-based vaccines could have a great impact to fight this virus, since mRNA vaccines are faster and cheaper to produce than conventional vaccines. In addition, this process can be standardized and scaled to allow a quick response to this pandemic. Besides, an mRNA-based vaccine is safer both for the personal that produces it and for the patient, since it does not use highly infectious viruses. Taken altogether, we suggest a strategy (Figure 5) to develop an mRNA-based vaccine against SARS-CoV-2 by targeting the spike S protein (S protein) which binds to membrane ACE2 (angiotensin-converting enzyme 2) receptors. ACE2 receptors can be found in several organs such as lungs, heart, kidneys and the gastrointestinal tract, which are the SARS-CoV-2 targets. We could expect that once the mRNA encoding the S protein is delivered into the myocytes of the muscle, it will be translated and secreted and taken up by macrophages, dendritic cells and other cells of the immune system to be presented to T and B cells. Once a B cell response is mounted, the resulting antibodies could block the binding of the virus to the ACE2 receptors and thus prevent the virus to enter to the target cells. In addition, activated T cytotoxic cells can help to destroy the invading virus (Figure 5).

Messenger RNA can be used in Biotechnology and Medicine as RNA vaccines against infectious diseases and also in cancer immunotherapy to deliver multiple antigens with one immunization [13,14]. RNA vaccines might have several advantages over DNA-based vaccines, since they only need to reach the cytoplasm, are more immunogenic and do not possesses oncogenic potential via integration into the host genome [13,14].

Typically, RNA-based vaccine consists in a functional mRNA encoding an antigen, which is delivered into the cellular cytoplasm of target cells where it can be translated via poly-some formation [15]. There are different non-viral strategies for the delivery of mRNA-based vaccines, including naked mRNA vaccines, gene gun delivery method, protamine condensation of the mRNA, adjuvants and encapsulated mRNA-based vaccines. The strategy of using naked mRNA vaccines uses only mRNA formulated in a buffer and injected directly into the individual [16]. The gene gun delivery is an alternative method of mRNA delivery directly into the cell cytoplasm [17]. Specifically, it is a nanotechnology method in which the mRNA is coated onto gold particles, which are then accelerated toward a stopping plate by a helium pulse and penetrated into the cytoplasm of target cells [17]. Another method to improve mRNA stability is incubating it with protamine and inject this protamine-protected mRNA into the individual [18]. Additionally, the mRNA can be injected with molecules such us poly I:C RNA and CpG- containing motif molecules [19]. Lastly, another method, based on nanotechnology, has been used to deliver mRNA-based vaccines. Nanoparticles of cationic liposomes have been used to protect and deliver mRNA vaccines into the individuals [20,21].

The mRNA-based vaccines can be used to immunize against infectious diseases and several types of cancer [22]. Immunization against the Zika virus and against the rabies virus with mRNA-based vaccines are only two cases where this immunization procedure shows promising results [22,23,24]. Both mRNA-based vaccines were delivered by using nanoparticles of cationic liposomes in mice. The vaccines provided protection against lethal doses of virus in those immunized mice [22,23,24]. On the other hand, mRNA-based vaccines have been used to immunize against several types of cancer [13,14]. This strategy relies on the generation of a host immune response against specific tumor-associated antigens by the cytotoxic T cells, which are able to recognize and kill tumor cells. Once the mRNA-based vaccine is delivered into the cytoplasm of antigen presenting cells (mainly macrophages), the resulting antigen is presented together with MHC class I molecules to CD4+ T cells, activating a cellular response that leads to the destruction of the tumor cells. Additionally, the mRNA can act as an adjuvant and provide costimulatory signals via toll-like receptors. To illustrate this strategy, we describe two examples. An mRNA-based vaccine against malignant melanoma was developed using the injection of naked mRNA encoding melanoma-associated antigen [25]. In addition, another mRNA-based vaccine was developed against triple-negative breast cancer by delivering mRNAs encoding tumor-associated antigens into cationic liposomes [26].

## 4. Other Uses of mRNA as Nanomedicines

The mRNA also has other applications in nanomedicine, including protein replacement, gene editing, regenerative medicine and monoclonal antibody production. The application of mRNA for protein replacement therapies is intended for the supplementation of proteins that are not expressed or not functional and also for the expression of foreign proteins that are able to activate or inhibit cellular pathways. These therapies have been mainly used to target diseases in organs such heart, lung, and liver. For example, diseases such as hemophilia B, cystic fibrosis or Fabry disease are the subject of clinical trials [27,28,29]. On the other hand, gene editing is a new therapeutic option to treat a variety of genetic diseases and this technology uses programmable nucleases, which are able to perform a double stranded break at specific target locations of the genome in the presence of a guide RNA that directs the nuclease to the target sequences (CRISPR/Cas9). The gene editing nuclease can be delivered, as the mRNA form together with the guide RNA, and it has achieved greater editing efficiency [30]. The goal of regenerative medicine is to replace or repair cells which have been injured or lost and restore the normal function of damaged organs. The regeneration process requires functional proteins including cytokines, transcription and growth factors which control cell growth, cell differentiation, and cell migration, which can be delivered in the mRNA form into cells. One example of this is the generation of insulin secreting B-cells for type 1 diabetes [31]. In addition, passive immunization with mRNA encoding monoclonal antibodies is gaining great biomedical interest. In small rodents, the injection of mRNA encoding monoclonal antibodies have demonstrated the production of antibody titers [32].

## 5. Basic Elements to Design an mRNA-Based Vaccine

A keen awareness of mRNA biology is vital at the time of developing an mRNA-based vaccine. Most, if not all, of eukaryotic mRNAs are composed of a coding region (ORF) flanked by a 5′ and 3′ untranslated regions (UTR), a 5′ 7-methylguanosine triphosphate (m7G) cap and also a 3′poly (A) tail (Figure 6). It would also be important to back-translate the amino acid sequence of the ORF to DNA sequence and use the optimal codon usage and chemically synthetize a DNA template to produce an mRNA which could be more stable and a better template for translation. All of those elements are critical for mRNA stability and translation. Although, the poly (A) tail and the UTR can be included into the DNA template used for transcription, the m7G must be capped with cap analogues, which is not 100% efficient, and thus a portion of the mRNA transcripts are not capped at all [21]. The resulting uncapped mRNA cannot be efficiently translated and thus the mRNA serving as templates for protein translation is much less. To circumvent this, we would suggest an enzymatic approach using the human methyltransferase and capping enzymes (Figure 7), both of them first cloned and expressed in our laboratories [33,34]. However, both enzymes have not been expressed in scale for commercial purposes, thus hampering this approach.

## 6. Advantages of Using Eukaryotic RNAPII to Synthetize mRNA

Usually, the mRNA is synthetized “in vitro” using bacteriophage RNA polymerases, which are highly actives. However, these RNA polymerases can only synthetize an mRNA up to 5 Kb long. Eukaryotic RNAPII is able to synthetize mRNAs over 50 Kb or longer. In addition, RNAPII is less error prone than bacteriophage RNA polymerases, which is important to avoid mutations or the creation of stop codons.

By the present, all TFIIs necessary for transcription are cloned and expressed and it is possible to purify core RNAPII, thus allowing us synthesize mRNA in a test tube with all proteins of eukaryotic origin. Recently, Fujiwara and Murakami have reported a method to assemble large amounts of PIC using purified TFIIs and RNAPII from budding yeast [35]. Those PICs were assembled and purified further using glycerol gradient sedimentation [35]. We have also been attempting to assemble PICs from fission yeast by using whole cell extracts, which contain a full set of TFIIs, RNAPII and regulatory factors (Maldonado, E., unpublished results). For those experiments we used a DNA template containing a super core promoter (SCP) based on the work of Gershon and collaborators [36] and we added a GAL4 binding site upstream of the TATA box to allow strong in vitro transcription activation by the powerful GAL4-VP16 chimeric transcription factor. Those PICs have been assembled and further purified by the use of glycerol gradient sedimentation (Figure 7). The isolated PICs are highly active in transcription initiation and elongation and RNAs of more than 2 Kb long can be obtained. We think that by the use of similar methods it is possible to obtain large amounts of in vitro transcribed mRNA using eukaryotic RNAPII. Moreover, eukaryotic methyltransferase and capping enzymes can be added to the in vitro reaction together with precursors for the m7G. This point is very important since it has demonstrated that capping is a co-transcriptional process [33,34].

In the near future, we could image that a multitalented recombinant polypeptide of eukaryotic origin would be able to recognize the promoter in the DNA template and perform the elongation of the mRNA chain. To engineer this multitalented polypeptide, we must know all fundamental and necessary domains in each of the individual subunits of core RNAPII and TFIIs involved in the recognition of the promoter and in the function of transcription elongation. Moreover, another polypeptide containing methyltransferase and capping activities could be engineered to add the m7G cap on all the mRNAs to produce a template 100% efficient for translation. This would be an important advance on protein biopolymers able to synthesize mRNA for uses in mRNA-based vaccines.

## 7. Conclusions

In this review, we have described the PIC formation process on eukaryotic TATA-containing promoters and additionally the PIC complex formation on two eukaryotic non-TATA promoters, such as HomolD box and INR-containing promoters, in which the mechanisms of PIC formation are not fully understood yet.

A strategy is outlined to develop an mRNA-based vaccine to fight the SARS-CoV-2 virus, which is causing the devasting COVID-19 disease at the present-days. In addition, we reviewed the main applications of the mRNA-based nanomedicines and their potential therapeutic uses. Some of the mRNA-based vaccines have moved forward to clinical trials. In the case any mRNA-based vaccines were successful; these processes will be streamlined to establish a large scale-production platform and in near future mRNA vaccines will be used in humans and animals as well.

The main problems facing in vitro transcription of mRNA is the 5′ capping of the transcribed mRNA and the size of the transcript that can synthetize a viral RNAP. We have suggested an in vitro system using eukaryotic RNAPII, which is able to synthetize longer transcripts of up to 50 Kb long. The in vitro transcription setup is highly active and includes the capping system, which can synthetize the m7G cap in a co-transcriptional manner and makes the whole process more effective. Since, the in vitro transcription system used is rather simple, the functional mRNA product can be easily purified and used as a vaccine.

## Figures and Tables

**Figure 1 polymers-12-01633-f001:**
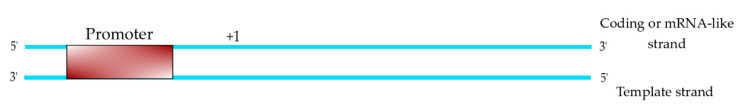
Description of the DNA template. The top strand is the coding strand or RNA-like strand, while the bottom strand is the template strand. +1 is the transcription start site.

**Figure 2 polymers-12-01633-f002:**
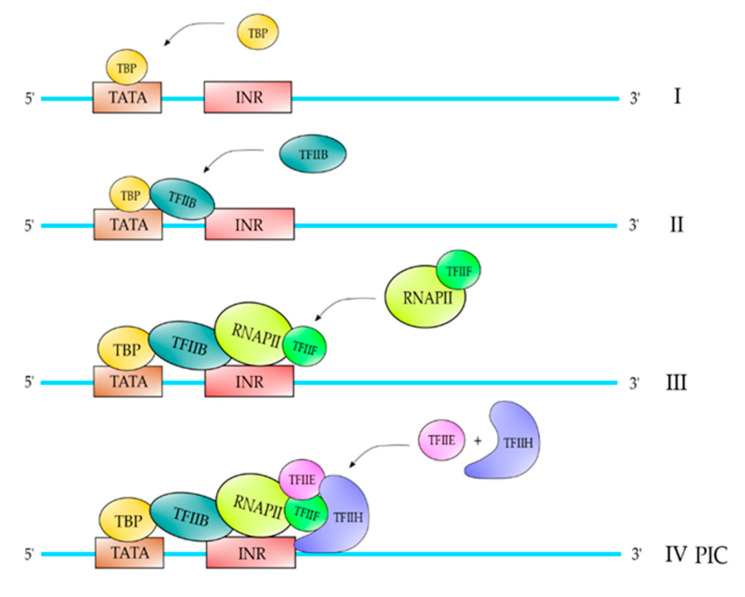
PIC formation on TATA-containing promoters. The first step of PIC formation on this kind of promoters is the binding of TBP to the TATA element (I), which follows the binding of TFIIB through interactions with TBP and the promoter near to the transcription start site (II). After the binding of TFIIB, the complex RNAPII-TFIIF is assembled into the complex (III), which follows the binding of TFIIE and TFIIH to form a PIC on the promoter (IV). Only the top strand of the promoter is shown in the figure for didactic purposes, but the PIC is formed on the double stranded promoter. This is also valid for Figure 3 and Figure 4.

**Figure 3 polymers-12-01633-f003:**
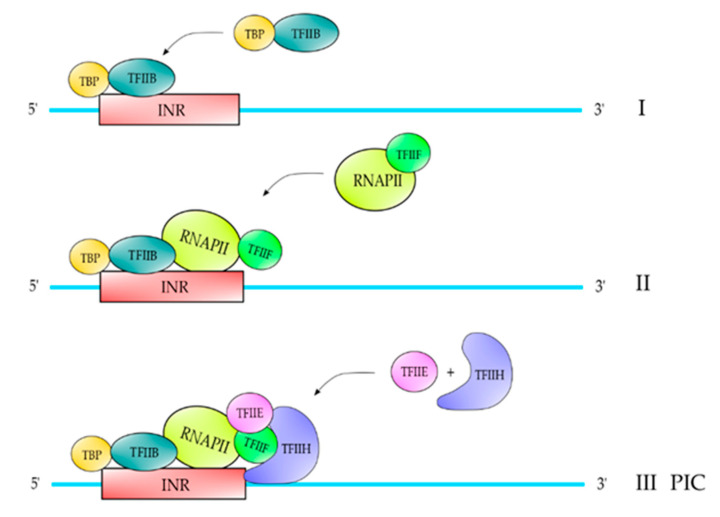
PIC formation on an INR-containing promoter. In those promoters that contain an INR, the complex TBP-TFIIB binds to this element (I) and after the complex RNAPII-TFIIF enters the complex (II), which follows the binding of TFIIE and TFIIH to complete the PIC formation (III). This model of PIC formation has not been published yet, but we have determined this pathway using pure RNAPII and recombinant factors from fission yeast and the nmt1 promoter in which the TATA box was deleted.

**Figure 4 polymers-12-01633-f004:**
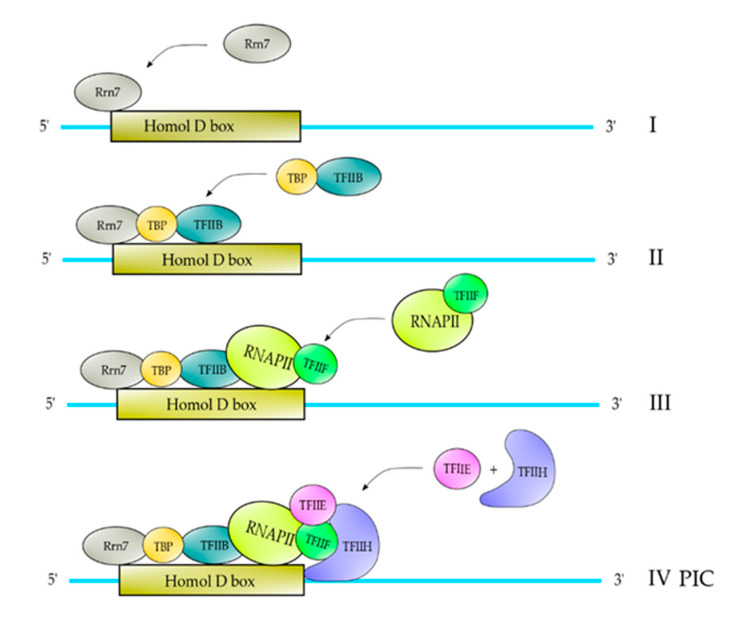
PIC formation on HomolD-box-containing promoters. In the first step of PIC formation, the transcription factor Rrn7 binds to the HomolD-box (I), which in turn is able to recruit the TBP-TFIIB complex (II). After TBP-TFIIB are recruited to the complex, it follows the binding of RNAPII-TFIIF (III), and then the TFIIE and TFIIH are recruited to the complex to form the complete PIC (IV).

**Figure 5 polymers-12-01633-f005:**
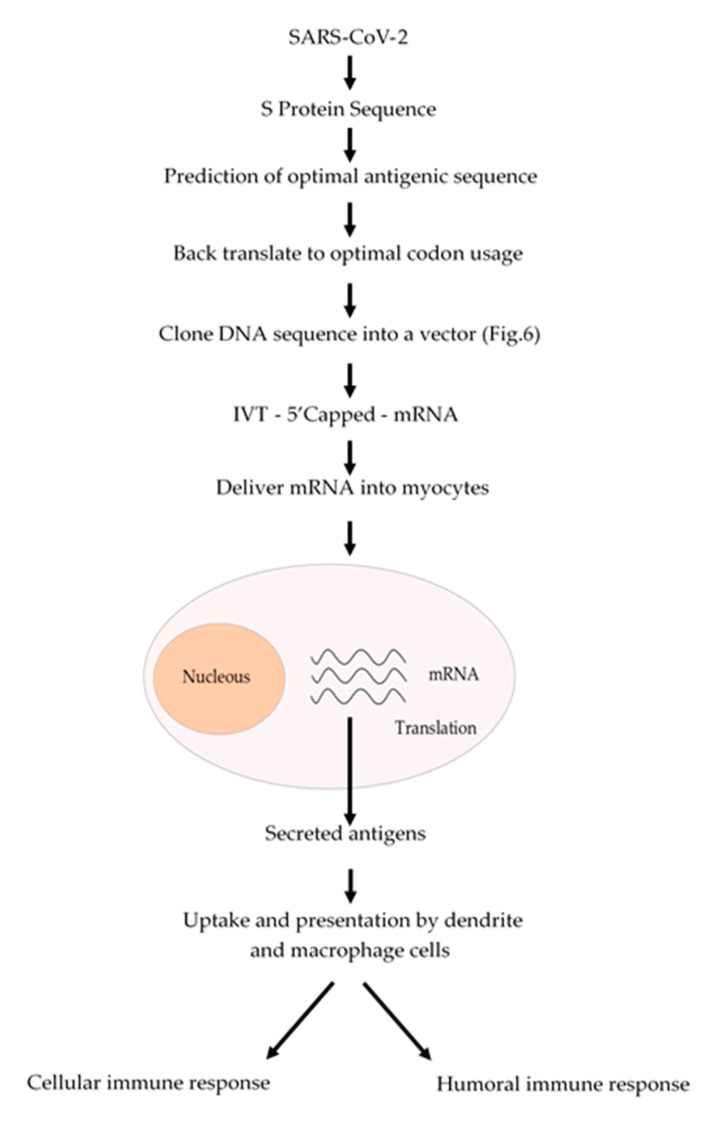
Schematic procedure to produce an mRNA vaccine against SARS-CoV-2. From the analysis of the amino acid sequence of the S protein an antigenic region can be predicted. Whenever possible, a region of the S protein which is less prone to mutations should be selected. The amino acid sequence of the selected region can be back translated to DNA nucleotide sequence and it can be chemically synthetized according to the codon usage. Afterwards, this DNA sequence can be cloned into a DNA plasmid vector, as described in Figure 6, and used for in vitro transcription (IVT) by using a RNA polymerase to produce an mRNA that can be 5′ capped to be fully functional. This mRNA can be coated with lipid nanoparticles and delivered to the myocytes to be translated by the cell ribosomes. The resulting antigens will be secreted and take up by macrophages and dendritic cells, which will process and present the antigen to the immune system cells to activate both humoral (antibodies) and cellular (cytotoxic) immune responses.

**Figure 6 polymers-12-01633-f006:**
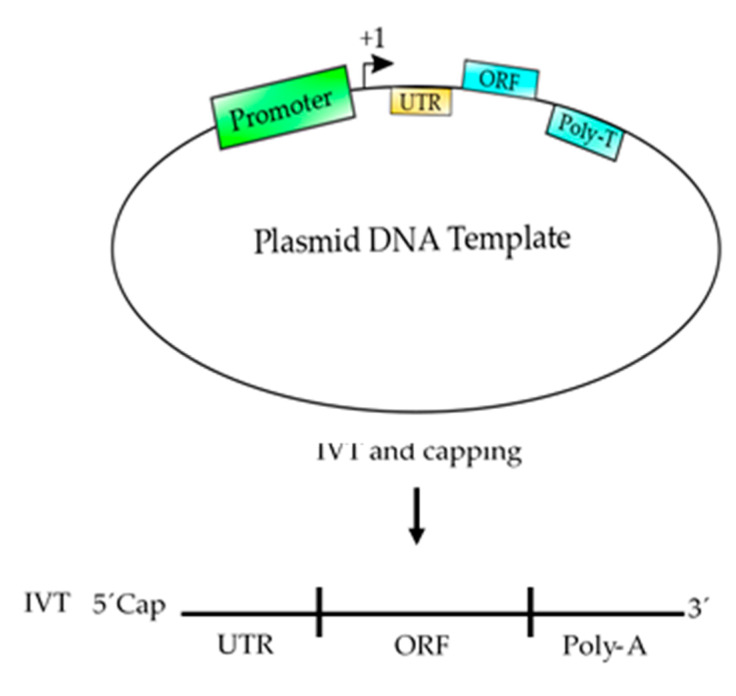
Schematic representation of a plasmid DNA template used to produce an mRNA-based vaccine. This template contains the basic elements to produce a functional mRNA. A promoter is needed for the RNA polymerase to start transcription at the +1 site. The 5′ UTR is added to enhance the translation and the stability of the mRNA and the ORF contains the coding sequence of the antigen, which could be synthetized according to the codon usage to enhance the translation process. Finally, a poly T of at least 200 nucleotides is needed to produce a poly A tail on the mRNA. After, the template is transcribed (in vitro transcription, IVT) using RNA polymerase and ribonucleotides to produce an mRNA, which should be 5′ capped to produce a functional mRNA to be used as vaccine.

**Figure 7 polymers-12-01633-f007:**
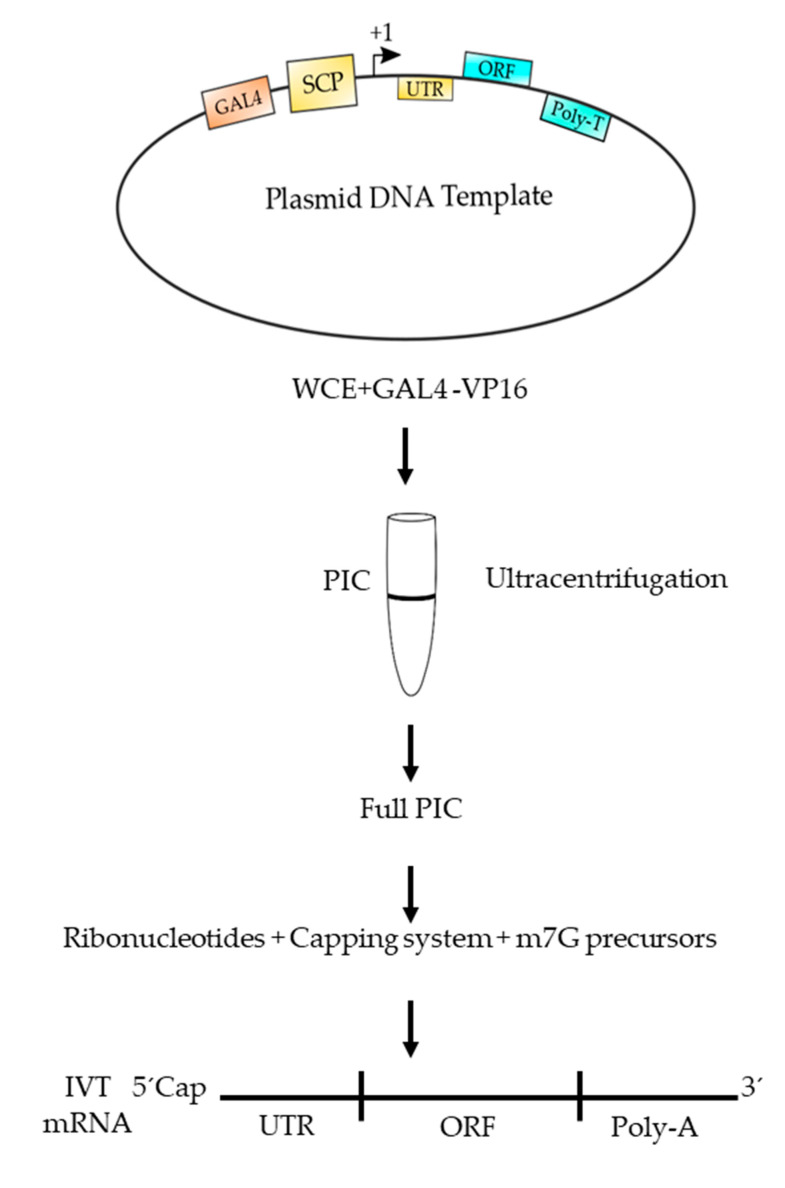
Outline of a procedure to produce a functional mRNA using eukaryotic RNAPII. The plasmid DNA template contains a super core promoter (SCP) and an upstream GAL4 binding site. Transcription starts at the +1 site, which is followed by an UTR (untranslated region) and an ORF containing the coding region to produce the polypeptide chain. A poly T tail is included in the template to produce a poly adenylated mRNA. The template is mixed with a highly active whole cell extract (WCE) from fission yeast, plus GAL4-VP16 activator and subjected to ultracentrifugation through a glycerol gradient (5–15%). Afterwards, those templates containing a fully assembled PIC are selected and used for transcription by adding ribonucleotides, together with methyltransferase and capping enzymes (capping system) plus the necessary precursors to form a m7G cap. The produced mRNA can be used as an mRNA vaccine.

**Table 1 polymers-12-01633-t001:** RNAPII, TFIIs (GTFs) and mediators.

TFII	Polypeptide Composition	Function
TBP	1	Recognize TATA elements and INR
TFIIB	1	Bridges TBP and RNAPII-TFIIF
TFIIF	3 ^a^	Helps to recruit RNAPII to the PIC
TFIIE	2	Stabilizes RNAPII-TFIIF to the PIC
TFIIH	10	Kinase and helicase activities
Core RNAPII	12–14	Synthesis of the mRNA
Mediator	24 ^b^	Target of regulatory factors Bridge the PIC with activator factors
Rrn7	1	In *S. pombe*, this transcription factor recognizes the HomolD-box of RPG promoters

^a^ Budding yeast has an extra subunit, which is not essential. In most of the eukaryotic organisms studied, TFIIF has only two subunits; ^b^ The mediator from budding yeast has 24 subunits, however, the mediator from other organisms might have more than 24 subunits.

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
