# Peer review of "Enzymatic Protein Biopolymers as a Tool to Synthetize Eukaryotic Messenger Ribonucleic Acid (mRNA) with Uses in Vaccination, Immunotherapy and Nanotechnology"

_polymers, 2020, doi:10.3390/polym12081633_

Round 1
Reviewer 1 Report
The author has well addressed the comments and I suggest to publish in Polymers.
Author Response
Thank you very much for your expert opinion.
Reviewer 2 Report
The work conducted by Urbina et al explores the polymerase II (RNAPII) and the pathways necessary for a preinitiation complex formation. They introduced different methodologies to transcribed mRNA in vitro for the production of vaccines to fight infectous diseases, to be used in immunotherapy treatments or in the delivery of mRNA to proteins present within the cells. Even though these pathways have been revised previously, the authors presented a very interesting take by tackling their importance to the current pandemic situation the entire world is facing. It was a very intersting and scientifically sound manuscript, with well organized ideas and a clear message.
The only point that I would raise is the absence of a conclusions sections. The authors present an interesting paragraph on what we should expect from the future, but it would be important for them to have presented one or two paragraphs focusing on the most essential points of the mansucript and what we should retain from their review. I would recommend that addition prior to publication.
Author Response
Thank you very much for your suggestion of including a short paragraph on conclusions. We have included the main conclusions of this work. We hope it will improve the manuscript content. Should we note that this manuscript also describes the preinitiation complex formation on eukaryotic promoters other than tata-box. this is novel, since nobody has described a mechanism to explain complex formation on those promoters, which are the majority in mammals.
This manuscript is a resubmission of an earlier submission. The following is a list of the peer review reports and author responses from that submission.
Round 1
Reviewer 1 Report
Please find attached.

Reviewer 2 Report
This paper tried to summarize the progress of protein biopolymers that can be utilized for synthesizing mRNA for uses in mRNA-based vaccines against cancer cells. However, probably due to the limited language, the idea of the authors could not be delivered successfully to readers. The abstract and whole paper have many grammatic mistakes, and the outline and purpose for this paper is not clear at all. The references are not updated, with only 4 out of 30 are from 2019, and lots of papers are from 10 years ago. Therefore, this review work is not suitable for publication until the quality could be improved dramatically.